# Investigation into the Physicochemical and Textural Properties of an Iron-Rich 3D-Printed Hybrid Food

**DOI:** 10.3390/foods12071375

**Published:** 2023-03-23

**Authors:** Coline Schiell, Stéphane Portanguen, Valérie Scislowski, Thierry Astruc, Pierre-Sylvain Mirade

**Affiliations:** 1ADIV (Association pour le Développement de l’Institut de la Viande), 63039 Clermont-Ferrand, France; 2Université Clermont Auvergne, INRAE, UR370 Qualité des Produits Animaux (QuaPA), 63122 Saint-Genès Champanelle, France

**Keywords:** food design, iron deficiency, 3D food printing, personalized nutrition, liver, red lentil

## Abstract

In the context of dietary transition, blending animal-source protein with plant-source protein offers a promising way to exploit their nutritional complementarity. This study investigates the feasibility of formulating an iron-rich hybrid food product blending plant-source and animal-source protein ingredients for iron-deficient populations. Using a commercial 3D-food printer, two different-shaped products composed mainly of pork and chicken liver and red lentils were designed. After baking at 180 °C with 70% steam, the 3D-printed products were packed under two different modified atmospheres (MAP): O_2_-MAP (70% oxygen + 30% carbon dioxide) and N_2_-MAP (70% nitrogen + 30% carbon dioxide) and stored at 4 °C. pH, water content, a_w_, lipid oxidation, heme iron and non-heme iron contents and textural properties were measured after 0, 7, 14 and 21 days in storage. After 21 days in storage, the 3D-printed hybrid products had an iron content of around 13 mg/100 g, regardless of the product form and packaging method. However, O_2_-MAP products showed significant (*p* < 0.05) time–course changes from day 0 to day 7, i.e., an increase in lipid oxidation, a decrease in heme iron content and an increase in product hardness, gumminess and chewiness. This work opens prospects for developing hybrid food products that upvalue animal by-products.

## 1. Introduction

Anemia is a global health threat, with about 1.62 billion sufferers worldwide, i.e., nearly 25% of the world’s population [1]. Anemia mainly affects preschool-aged children and pregnant women but is also associated with a sedentary lifestyle and is more prevalent in developing nations [1]. Iron has many metabolic functions. It is essential for oxygen transport in the body, as it complexes with the heme of hemoglobin and myoglobin. Dietary iron can be found in different forms: highly bioavailable heme iron (HI), which is bound to protein and only found in animal-source proteins, and free non-heme iron (NHI), which is found in both in plant-source and animal-source proteins but has lower bioavailability, as it is more affected by other diet factors. Indeed, anti-nutritional factors such as tannins and phytates can bind the iron and reduce its bioavailability. The essential HI only accounts for 10–15% of total dietary iron intake but represents about two-thirds of total absorbed iron [2].

Iron deficiencies can be avoided via various approaches, such as iron fortification and iron supplementation. In this study, iron is provided by a natural route by upvaluing HI-rich animal by-products such as liver [3] in combination with lentils that do not contain highly assimilable iron but carry other nutritional and technological advantages such as dietary fiber or vitamins and minerals. NHI is the majority form of iron found in liver (77–86% of total iron) due to its role in iron storage. Indeed, liver iron is mainly associated with ferritin and insoluble hemosiderin, which act as iron storage compounds. The remaining HI is present in hemoproteins such as hemoglobin [4]. Unlike animal matrices, plant matrices contain NHI in soluble forms such as ferrous ions (Fe II), ferric ions (Fe III) and complexed iron, as well as insoluble forms. Only a few studies have addressed the soluble iron content of lentils. According to Quinteros et al. [5], soluble iron represents 6.7–8% of total iron after cooking, breaking down as 0.4–1% of Fe (II) and 6.2–7% of Fe (III). Most of the iron found in lentils would therefore be in the form of insoluble forms and complexed iron such as phytoferritin.

In addition to its nutritional efficiency, offal may increase the bioavailability of NHI present in plant-source food materials [4]. Indeed, the bioavailability of iron in the organism depends not only on the amount of iron ingested but also on the form of the iron and the composition of the food matrix eaten (dietary factors). Therefore, although plant NHI is known to have a lower bioavailability compared to HI, studies have shown that animal protein degradation products enhance the availability of both HI and NHI [6], in a process called the ‘meat-factor’. These protein compounds, which are also present in animal by-products [7,8], could have the capacity to bind NHI and thus improve solubility and bioavailability. This element supports the development of hybrid products that blend nutrients from animal and plant sources and thus offer the possibility of designing food products with a high nutritional content. This type of food is in line with current consumption trends and nutritional recommendations to rebalance the ratio of animal protein to plant protein towards more plant protein [9]. There has been a surge in studies demonstrating the nutritional value of pulses and the functional properties of their proteins, and which have encouraged industry and consumers to use them more. However, there has been very little research on blends on various protein sources and their ability to meet the nutritional needs of consumers, including specific populations.

In this context, 3D food printing (3DFP) is suitably tooled to deliver personalized nutrition through the development of functional foods [10]. Indeed, 3D printing (3DP) is an ‘additive manufacturing’ process that consists of depositing materials layer by layer based on a 3D computer model. The technology of 3DFP emerged in 2007 and made it possible to design customized foods from liquid, powder or paste materials, but it quickly demonstrated a potential for the rapid prototyping of formulations, making it equally possible to work on food design and texture [11]. There have been many studies on the applications of 3DFP technology for populations with specific needs, such as the elderly [12], young children [13] or people suffering from metabolic disease [14]. The 3DFP technology offers a wide range of ingredient choices and precise control of the process and quantities delivered, making it ideally geared to current diet demands and desires of consumers. 

The 3DFP technology therefore opens excellent opportunities to value-stream agrifood-industry by-products. One of the main advantages of 3DFP is that it can potentially increase the acceptability of a food by modifying its aesthetics to make it more attractive to consumers. The process can help to address global sustainability challenges by reducing food waste and costs. Recent studies have shown the feasibility of manufacturing high-value 3D-printed products from by-products, using porcine plasma proteins from slaughterhouse blood, soy and pea proteins from industrial extractions or surpluses [15], grape pomace and broken wheat [16], salmon skin gelatin [17], or even more exotic products such as protein-rich cricket powder [18]. The by-products usually contain nutrients of interest such as proteins, fiber, antioxidants or essential minerals [19]. Nevertheless, in most cases, the 3D-printed products are made from powders via processes that mix all the ingredients and strongly denature the structure of the raw materials. Indeed, the main challenge for 3D-printed products remains the printability of the raw materials in relation to their texture and therefore their composition, which can be highly variable. 

The present work investigates the feasibility of formulating an iron-rich hybrid food product for iron-deficient populations and the time-course stability of the hybrid food in order to verify that it provides the necessary quantity of available iron over the full length of time in storage. This is achieved using physicochemical and textural analyses to characterize the properties of printable food materials and 3D-printed food products.

## 2. Materials and Methods

### 2.1. Preparation of Printable Food Matrices

As indicated in Figure 1, the first step of the experiment was to design a hybrid food containing an animal part and a plant part. These two parts were prepared separately before being assembled together during the printing process.

Animal part. Raw poultry liver (Sedivol, Isserteaux, France) and raw pork liver (Limoujoux Auvergne Viandes et Salaisons, Clermont-Ferrand, France) were coarsely chopped, then minced with a 3 mm grinder (Hobard 4732, Troy, USA) and finely ground to 0.2 mm with a grinder (Stephan Mikrocut, Hameln, Germany). To obtain the final mixture, the two livers were mixed together with raspberry vinegar (Clovis, Reims, France) and salt (Table 1) in a blender equipped with a flat beater (Kenwood Chef Classic KM 400/410, Havant, UK) to meet a traditional process. To improve printability, the animal-source mixture was pre-cooked in cooking bags (200 g of product per bag) in a water bath at 50 °C for 15 min in order to increase its viscosity without completely denaturing the proteins [20]. The bags were then cooled by immersion in melting ice. Pork liver and poultry liver were selected primarily for their high iron content. Based on preliminary rheological analyses (data not shown), pork liver had the highest viscosity and was more suitable for 3D printing. Nevertheless, from an organoleptic point of view, pork liver has a more pronounced flavor. Therefore, the formulation of the animal part was optimized to about 70% poultry liver and 30% pork liver.

Plant part. Dry red lentils (Lens culinaris L.) (Sabarot, Chaspuzac, France) were rinsed with clean water to remove impurities, then cooked for 15 min in unsalted boiling water (without pre-soaking) at a ratio of 1:5 (*w*/*w*). The cooked lentils were then drained, rinsed in cold water and blended (Blixer^®^ 4 V.V, Robot-Coupe, Montceau-les-Mines, France). The lentil puree was then sieved (particle size: 1.40 mm) to prevent large particles from clogging the nozzle during printing. For technological, organoleptic and nutritional reasons, lupin flour (Markal, Saint-Marcel-les-Valence, France), peanut oil (Scamark, Ivry-sur-Seine, France), curry powder and salt were added to the lentil puree in the blender with a flat beater (Kenwood Chef Classic KM 400/410, Havant, UK) (Table 1).

The animal and plant parts were placed in plastic bags and stored in a refrigerator overnight at 4 °C. A printable recipe for each part (animal-source and plant-source) was determined based on preliminary tests (see Table 1 for details on recipe composition). 

### 2.2. Printing Process

For these experiments, the food matrices were 3D printed using the commercial extrusion-based ‘Foodini’ 3D printer (Natural Machines, Barcelona, Spain). Each formulation was calibrated using the auto-calibration Foodini tool to define the optimal printing parameters (Table 2). The food matrices were printed using a 4 mm circular nozzle to accommodate their particle size.

For 3D printing, two forms were designed with different geometries and varying proportions of the animal-source and plant-source mixtures (Figure 2). The first form (C) resembles a filled cookie with the offal mixture hidden inside, corresponding to an animal/plant ratio of about 20:80. The second form (F) looks like a flower with petals filled with the offal mixture, corresponding to an animal/plant ratio of about 15:85. These two different geometrical forms were chosen with the idea that the location of the offal (in direct contact or not with oxygen) could have an impact on the organoleptic qualities of the product during storage.

### 2.3. Post-Printing Process and Modified Atmosphere Packaging (MAP)

Post-processing is often necessary after 3D printing to stabilize the product in terms of structure and food safety. In the present study, the objective was to find a compromise in terms of heating in order to fix and maintain the shape of the 3D-printed food, while ensuring a pleasant texture and good microbiological safety [21]. To reach a core temperature of 72 °C, forms C and F were baked at 180 °C with 70% steam for 5 and 2.75 min, respectively, using an electric oven (HMI Thirode, Emeraude III+, Mitry-Mory, France). The baking rate was determined by measuring the core temperature of the 3D-printed food in the oven. A calibrated K-type thermocouple probe (0.6 mm diameter; accuracy ± 0.5 °C) was inserted, rigorously fixing the height into the food, and connected to a data acquisition system.

After baking then cooling down to room temperature, the 3D-printed products were packed in plastic trays (14.5 × 21.5 × 5 cm) using a sealing machine (Multivac C500, Wolfertschwenden, Germany) under a controlled atmosphere and stored at 4 °C. To compare the impact of storage conditions on product properties, the 3D-printed products were packed under two different MAP conditions: O_2_-MAP with 70% O_2_ + 30% CO_2_ and N_2_-MAP with 70% N_2_ + 30% CO_2_. These two MAP conditions were chosen as they are widely used by industry for meat products: an O_2_-rich atmosphere allows for optimal coloring of red meat, and inert nitrogen gas limits the oxidation of lipids and pigments [22].

N_2_, CO_2_ and O_2_ levels were measured in the trays during storage using a gas analyzer (Gaspace Advance, Nanterre, France).

### 2.4. Characterization of the Properties of the Raw Food Materials and 3D-Printed Products

Physicochemical analyses were performed on both the raw materials and the 3D-printed foods. To study the properties of the end product, the entire 3D-printed product was milled in a blender (Waring, Commercial Blender, USA), and 3 samples were taken from the grind (in triplicate) for each subsequent analysis.

#### 2.4.1. Determination of Water Content, Water Activity and pH

Water content was quantified by weighing the dried matter after 24 h at 105 °C in a drying oven (ED240 Binder Gmbh, Tuttlingen, Germany). The water content X, expressed in kg water/kg dry matter, was calculated as follows (Equation (1)):(1)$$X=\frac{\mathrm{mi}-\mathrm{mf}}{\mathrm{mf}}$$
where mi is initial mass (kg) of the analyte sample, and mf is final mass (kg) of the sample after drying.

Water activity (a_w_) was determined at 20 °C with an a_w_ meter (Novasina TH-500, Lachen, Switzerland). Prior to each series of analyses, the meter was calibrated using certified standards with the following a_w_ values: 0.11; 0.33; 0.53; 0.75; 0.90; and 0.98. For each measurement, between 2 and 5 g of sample were placed in the measuring cell. The equilibrium state was checked using Novasina Novalog software (version 01.12), and the a_w_ value recorded corresponded to the extension of the asymptote of the curve on the *y*-axis. 

For pH determination, 0.5 g of sample was dissolved in 5 mL of distilled water (1:10 dilution) and thoroughly homogenized using a Polytron PT2100 dispersing unit (Kinematica, Malters, Switzerland). pH was measured using an Inlab 427 probe previously calibrated with standard solutions of pH 4 and pH 7 and connected to a pH-meter MA235 (Mettler-Toledo, Columbus, OH, USA).

#### 2.4.2. Iron Content Measurement

The two chemical forms of iron, HI and NHI, were measured in order to track their time–course in the product and as a function of MAP conditions. 

HI content was determined by the Hornsey method [23]. Briefly, 5 g of ground sample was extracted in an acidified acetone mixture (acetone/water/concentrated HCl: 20/4.5/0.5). The mixture was then stirred, incubated in the dark at room temperature for 24 h and filtered through Whatman filter paper. The absorbance of the mixture was read at 640 nm using a UV–Visible spectrophotometer (UVisco V-1800, LC-Instrument, Lisses, France). The HI concentration was calculated using a standard curve based on hydrochloride-hemin placed in acidified acetone mixture. 

NHI was estimated using ferrozine according to the method described in Ahn et al. (1993) [24]. Each sample (1 g) was ground with 5 mL of 0.1 M citrate–phosphate buffer (pH 5.5) and homogenized (Polytron PT2100, Kinematica, Malters, Switzerland) for 30 s at 13,500 rpm. Then, 1 mL of 2% ascorbic acid in 0.2 M HCl was added to 5 mL of the homogenized mixture and incubated for 15 min at room temperature. To precipitate the proteins, 2 mL of 11.3% trichloroacetic acid (TCA) and 120 µL of 0.39% NaNO_2_ were added and incubated for 20 h in a water bath at 65 °C. The mixture was then cooled to room temperature and centrifuged for 10 min at 20,000 g, and 2 mL of the supernatant was mixed with 0.8 mL of 10% ammonium acetate and 0.2 mL of ferrozine reagent. After filtration, absorbance was read at 562 nm against a blank. The NHI concentration was calculated using a standard curve plotted from FeCl_2_ placed in 0.1 N HCl solution. Total iron was calculated by summing HI and NHI content.

#### 2.4.3. Lipid Oxidation Measurement

The measurement of secondary compounds derived from the first oxidation changes gives reliable information on the oxidation state of meat products. Malondialdehyde (1,3-propanedial; MDA) is one of the most major aldehydes produced during the secondary oxidation of polyunsaturated fatty acids. The thiobarbituric acid reactive substances (TBARs) method is the main technique for quantifying all substances that react with the TBA, especially MDA, which is the main lipid oxidation compound [25]. The first step consists of acid extraction of the MDA contained in the product with trichloroacetic acid (TCA). For this, 2 g ± 0.01 g of ground sample was homogenized in 40 µL of a solution of 1 µg/µL BHT in ethanol, 80 µL of 0.1 M EDTA and 10 mL of 0.3 M TCA using a homogenizer at 8000 rpm (Polytron PT2100, Kinematica, Malters, Switzerland). The mixture was then centrifuged at 3000 g for 5 min, and the supernatant was filtered through a Whatman filter. The extract was then stored at 4 °C in the dark before being assayed.

For the reaction, 3 mL of the extract was collected and added with 3 mL of 55.5 mM TBA solution to produce a stained complex. The mixture was homogenized and incubated for 20 min in a water bath at 70 °C and then cooled for 30 min in melting ice. The maximum absorbance of this complex was read at 532 nm against a blank containing all reagents except lipids. Quantification was carried out using a UV–Visible spectrophotometer (UVisco V-1800, LC-Instruments, Lisses, France).

The result of the TBARs index is calculated as follows (Equation (2)):(2)$$\mathrm{TBARs}\;(\mathrm{mg}\;\mathrm{MDA}/\mathrm{kg})=\frac{0.72}{1.56}\times \frac{A532\times \;\mathrm{Vs}\;\times \;\mathrm{Vf}}{W}=A532\times 4.67$$
where A532 is absorbance of the assay solution at 532 nm, W is the weight of the sample (g), Vs is the volume of the TCA + BHT + EDTA dilution solution (mL), Vf is the volume of filtrate collected, and 0.72/1.56 corresponds to the molecular extinction coefficient of the TBA–MDA complex [26].

#### 2.4.4. Texture Profile Analysis (TPA) of the 3D-Printed Products

The instrumental texture properties of each 3D-printed product were evaluated at different storage timepoints through storage and under different MAP conditions, using an EZ-Test LX texture analyzer (Shimadzu, Noisiel, France). The Texture Profile Analysis (TPA) test was performed with double compression at 50% sample height using a 50 mm-diameter cylinder probe at a displacement speed of 20 mm/sec and a trigger force of 0.5 N. The tests were carried out at 20 °C, and the TPA data were collected using TRAPEZIUM-X software (version 1.5.5). Due to its geometry and flat top surface that facilitate this type of measurement, TPA was performed only on form C. For each product, 3 cylindrical samples (⌀20 mm × 16 mm) were cut across the height of the ‘cookie’ with a cookie cutter.

Hardness (N), springiness, cohesiveness, gumminess (N) and chewiness (N) were used to characterize the textural properties of the 3D-printed products.

### 2.5. Statistical Analysis

Three independent repetitions were used to calculate the means and standard error of the mean (SEM). Multivariate analysis of variance (ANOVA) was performed on the data to determine statistically significant differences between samples in terms of form, MAP conditions and time in storage. In addition, statistically homogeneous groups using the Tukey HSD post hoc test were identified by pairwise comparisons between the means of each group. 

Principal component analysis (PCA) was performed using product form (C or F), time in storage (D0, D7, D14 and D21) and MAP conditions (N_2_-MAP or O_2_-MAP) as explanatory variables to highlight the variables that had the greatest impact on the properties of the 3D-printed product. As the variables have unequal units and variances, the data were centered and reduced.

Analytical results were processed using STATISTICA software version 13.3 (TIBCO Software Inc., Palo Alto, USA), and the threshold for significance was set at *p* < 0.05 for all analyses.

## 3. Results and Discussion

### 3.1. Nutritional Values of 3D-Printed Products

Table 3 presents the estimated nutritional profile of the 3D-printed products based on the values of the CIQUAL (French food composition table) database [27]. The 3D-printed products contained 14.9–15.5 g protein/100 g, which is equivalent to 8.9 g for the portion of C (19% of the recommended dietary allowance; RDA) and 4.8 g for the portion of F (10% of the RDA). The 3D-printed products were also high in dietary fiber, at 18% and 11% of the RDA for one portion of C and one portion of F, respectively, due to the addition of the plant-based source material. 

The detailed nutritional composition (Table 3) illustrates the complementarity between the two raw material sources in terms of carbohydrates, dietary fiber and protein. The development of new hybrid products is an opportunity to rebalance the diet and the ratio of animal to plant proteins. This type of product presents health benefits and is suitable for regular consumption.

### 3.2. Time–Course of Physicochemical Parameters

The physicochemical characteristics of the 3D-printed products were monitored for 21 days. Time–course changes in pH and water content during storage for each of the two food forms and each MAP condition are reported in Table 4.

Globally, for form C, pH increased by 0.11 to 0.12 pH units between D0 and D21 (*p* < 0.05) without a significant difference between the two MAP conditions. For form F, pH remained stable and then decreased very slightly (0.05–0.06 pH units) but significantly (*p* < 0.05) after day 14 for both storage conditions. As CO_2_ can dissolve in water and thus decrease pH, the equal concentration of CO_2_ in both conditioning modes may explain the lack of difference.. Although pH was initially lower for form C than for form F (6.28 vs. 6.41 at day 0; *p* < 0.05), there was no longer any significant difference between the two forms by day 7. This initial pH difference can be explained by the different compositions, as form C contained a higher proportion of the animal-source mixture, which had a more acidic pH (6.07) due to the presence of raspberry vinegar. 

a_w_ values showed no significant changes (*p* > 0.05), peaking at 0.98 throughout storage, regardless of product form and packaging method (data not shown). This suggests that MAP and time have no impact on this physicochemical parameter in the 3D-printed food. As a_w_ is high, the environment remained favorable to the development of microorganisms, which makes it necessary to store these products at 4 °C to slow down potential microorganism growth.

Form C demonstrated no significant change in water content with time in storage or type of packaging: there was a significant difference at day 7 between N_2_-MAP and O_2_-MAP (1.59 kg water/kg dry matter in N_2_-MAP versus 1.66 kg water/kg dry matter in O_2_-MAP, *p* < 0.05), but this difference disappeared afterwards. Once again, the differences observed between the two product forms (regardless of the time in storage, form F always contained less water than form C) are the result of their different initial animal-source/plant-source ratios.

### 3.3. Time–Course of Iron Content

The total iron concentration measured in raw chicken liver (Table 5) was two to three times higher than values reported in the literature, such as in Kongkachuichai et al. [28] (9.9 ± 0.8 mg/100 g) and in the CIQUAL database (9.0 mg/100 g on average) [27]. The total iron concentration in the raw pork liver was also higher than the value reported by Kongkachuichai et al. [28] (12.3 ± 3.5 mg/100 g), mainly due to a higher NHI content (9.4 ± 2.4 mg/100 g), but close to the values found in the CIQUAL database (18.4 mg/100 g on average) [27] and in Tomović et al. [29] (19.5–23.9 mg/100 g). This variability can be explained by various factors such as genetics, breeding, slaughter practices, feed or geographical location. Total iron concentration of the animal-source mixture is consistent with the proportions of liver used, i.e., 70% poultry liver and 30% pork liver.

For the plant part, the cooked lentil mixture had a total iron concentration of 8.76 ± 0.19 mg/100 g, which is much higher than that which was measured in the CIQUAL [27] for boiled red lentils (2.2 mg/100 g) and by Quinteros et al. [5] (2.01–2.25 mg/100 g depending on the cooking method). This discrepancy can be attributed to soaking the lentils, which leads to a decrease in NHI but was not performed here, and to the different cooking conditions. The plant-source mixture had a slightly higher iron content than the lentil mixture alone, because the lupin flour provides additional iron, as it has an iron content of 2.5–11.4 mg/100 g according to Eberl et al. [30].

Regarding the distribution of the iron forms, NHI was mainly observed in the liver-based preparations according to Bogunjoko et al. [31], whereas the plant mixtures did not contain HI but mainly free NHI (9.14 ± 0.18 mg NHI/100 g).

For the 3D-printed hybrid products, forms C and F contained 16.40 mg total iron/100 g and 15.77 mg total iron/100 g, respectively, on the day of fabrication (D0), i.e., a ratio of 94% NHI to 6% HI (Table 6 and Table 7). This initial composition reflects the proportions of the animal-source and plant-source mixtures incorporated in the recipes. After 21 days in storage, these 3D-printed products still had a total iron content of about 12–14 mg/100 g, regardless of product form (C or F) and packaging method (O_2_-MAP or N_2_-MAP), which would confer the right to use the nutrition claim “high in iron” as defined in the Annex to EU Directive 90/496/EEC on nutrition labeling for foodstuffs. 

This total iron content is naturally lower than that of some fortifying mixtures used as an ingredient based on pork liver (23.8 mg/100 g) [3] or beef lung powder (59.3 to 61.7 mg/100 g) [32], but higher than that of a conventional minced steak. These foods thus provide valuable fiber and polyunsaturated fatty acids (Table 1), but they are also well suited to the daily iron requirements of women and can thus help reduce the incidence of iron deficiency, as the RDAs for iron provided by the Institute of Medicine (IOM) [33] are 15 mg and 18 mg for women aged 14–18 and 19–50 years, respectively, increasingly to 27 mg in pregnancy.

In the context of a complex hybrid food, it is difficult at this stage to predict the bioavailability of iron in our products without further in vivo or in vitro studies. Rewashdeh et al. [34] studied the effects of dietary iron sources such as liver, lentils and liver plus lentil mixtures on iron bioavailability and concluded that liver-based diets can improve the bioavailability of plant-based iron, particularly when combined with lentils. Similarly, a recent study showed that some lentil proteins have the ability to bind to iron, which would decrease the free form of iron (Fe (II) and Fe (III)) prior to absorption and thus increase its bioavailability [35]. 

HI content decreased significantly during storage for both product forms independently of MAP, but a predominant effect of O_2_-MAP was observed. Indeed, HI content (5–10% of total iron) decreased more strongly when the products were stored under O_2_-MAP: between days 0 and 7, HI decreased by 0.65 mg/100 g under O_2_-MAP against 0.06 mg/100 g under N_2_-MAP. Regarding NHI, MAP conditions had no impact on NHI content, which decreased progressively over time for both product forms, although form F showed a significant difference between D0 and D21 (Table 7).

In contrast to what is presented here, some studies have shown that the NHI content of liver increases during storage in parallel with the decrease in HI content, due to degradation of the heme molecule that leads to the release of iron. However, Estévez et al. [36] showed that this increase occurred in the final stages of storage after 60 days. It is therefore possible that a 21-day follow-up is not sufficient to observe this conversion and the resulting increase in NHI. Furthermore, the amount of NHI is vastly higher than the amount of HI (about 15 to 43 times higher); the conversion of HI could possibly be masked by the natural decrease in NHI.

### 3.4. Evolution of Textural Properties

The TPA test is widely used to characterize the textural properties of food products. The double compression allows for the simulation of food chewing and thus brings the perceived texture of the food closer to its real acceptability by consumers [37]. 

Table 8 shows a significant effect of MAP conditions on product textural properties, especially hardness, gumminess and chewiness. The 3D-printed products had a significantly harder texture when kept in an oxygen-rich atmosphere, and this change in texture was detected from day 7 onwards and maintained until day 21. Textural changes appeared to have occurred between day 0 and day 7 mainly in the animal-source part, until a plateau was reached. Estévez et al. [38] observed a similar increase in the hardness of liver pâtés during refrigerated storage. Furthermore, Figure 3 shows that products stored under N_2_-MAP were more friable, crumbling more easily (Figure 3a), due to a marked dissociation of the animal and plant-based layers, in contrast to (Figure 3b) the denser and more cohesive products under O_2_-MAP that held together better. Even though they are slightly more brittle, ‘N_2_-MAP’ products were closer to the structure of the original food. These results are in agreement with Verma et al. [39], who found that pork loaves stored under N_2_-MAP showed significantly lower hardness than products kept under aerobic packaging throughout storage. This textural difference cannot be attributed to a variation in the water content of the product, as composition of the C form remained similar from D0 to D21 regardless of the packaging method. The main explanation of this textural change could be protein oxidations and interactions with other compositional components. Many studies converge to show that variation in the textural properties of meat products is related to changes in protein functionality and, in particular, interactions between the proteins, the continuous aqueous phase and dispersed fat. 

### 3.5. Lipid Oxidation

The TBARs method was used to determine lipid oxidation in the samples during storage under the two MAP conditions. As shown in Figure 4, the TBARs values of the 3D-printed products under O_2_-MAP increased significantly (*p* < 0.05) from 0.23 to 1.28 mg MDA/kg product for form C and from 0.09 to 0.67 mg MDA/kg product for form F between days 0 and 7, whereas TBARs values under N_2_-MAP remained globally constant at around 0.20 mg MDA/kg for both product forms during the entire 21-day period. Although there is no established threshold value in the literature, it is generally accepted that a product is oxidized if it contains more than 1 mg MDA/kg product [40]. This was the case for 3D-printed form-C products from day 7 onwards. After day 7, the TBARs values stabilized or decreased slightly. However, this does not mean that oxidation processes were no longer evolving. Indeed, studies have shown that TBARs values are a good indicator of the early stages of oxidation of a product, but that other methods, such as the determination of hydrosoluble Schiff base (HSB) values, are sometimes more appropriate for tracking lipid oxidation over time [41]. The values obtained here were close to those of Wang et al. [42] on chicken legs and Estévez et al. [36] on pork liver pâtés. Moreover, Del Olmo et al. [43] also observed that the N_2_/CO_2_ mixture limited the oxidation of pork meat products.

Lipid oxidation is known to cause a deterioration in product quality by affecting sensory properties such as color, flavor and texture (see Section 3.4). Lipid oxidation involves the degradation of polyunsaturated fatty acids and generates free radicals that lead to the deterioration of proteins and the oxidation of blood pigments [25]. Many studies have reported that lipid oxidation promotes the formation of metmyoglobin as a result of myoglobin oxidation and makes liver turn brownish grey during storage [36,44]. This change in liver color was also visible here in products stored under O_2_-MAP compared to N_2_-MAP (Figure 3b). In addition, the heme-protein oxidation caused by lipid oxidation also explained the decrease in measured HI content (Table 6 and Table 7). The difference in TBARs values between the two printed forms was mainly due to their different compositions and chiefly the ratio between the animal-source and plant-source parts. TBARs values were higher for form C, which contained more liver mixture and particularly chicken liver, which is rich in unsaturated fatty acids that are more prone to oxidation than saturated fatty acids. The animal part was also rich in NHI, which can further promote oxidation processes. The higher proportion of plant-source mixture in the F form could have helped to curb oxidation, as it contained antioxidant-rich ingredients such as curry powder [45].

### 3.6. Interactions between Physicochemical, Nutritional and Textural Parameters

Figure 5 shows the results of PCA performed to map the 3D-printed hybrid form C products to the main variables (physicochemical, nutritional and textural parameters) and explanatory variables (form, storage time and MAP). The additional variables correspond to N_2_, CO_2_ and O_2_ levels measured in the storage trays. These variables were not considered in the determination of the axes, but they serve to visualize the relationship of the variable with all the active variables in order to enrich the analysis. The results for the form F were not presented in the manuscript, because they are similar to those of the form C and provided less information (no texture parameter measurements).

This PCA specifically summarizes the link between the textural properties of the 3D-printed products and the physicochemical phenomena observed. Principal components 1 and 2 explained 69.9% of the total variance. The first component (about 51%) was more related to HI content, TBARs values and textural parameters as a function of MAP composition and thus explains the biochemical and textural changes related to level of product oxidation. Samples with low HI content were characterized by a harder, gummier and chewier texture and higher TBARs values. These samples correspond to those stored under O_2_-MAP. Component 2 explains less of the variance (about 19%) and is related to NHI content (the majority of the total iron content) and the adhesiveness of the product. Furthermore, physicochemical parameters such as pH, a_w_ and water content were fairly orthogonal to all the main PCA variables and were therefore not well represented on these axes, which is consistent with the non-significant results presented earlier. Nevertheless, the PCA map highlights the logical linkage between CO_2_ concentration in the tray and pH of the 3D-printed product, which confirms the hypothesis stated in Section 3.2. Similar results were observed with the biplot of principal components 1 and 3. Principal component 3 explained 11.1% of the total variance and was also related to NHI content, pH and adhesiveness of the product, but variables were poorly represented on the axis.

These results thus show that MAP composition has a major impact on the textural and physicochemical properties of the 3D-printed products and can separate the two groups of samples (O_2_-MAP and N_2_-MAP) (*p* < 0.05). Moreover, biochemical changes occurred mainly between D0 and D7, after which a plateau set in (non-significant differences between days 7, 14 and 21).

## 4. Conclusions

This study demonstrated the feasibility of producing an iron-rich hybrid 3D-printed product and showed the potential of 3DFP to drive rapid prototyping for the design of functional foods. 

The study extended our understanding of the effect of time in storage and MAP composition on oxidation mechanisms in hybrid products containing different forms of iron. The impact of these two factors, time in storage and MAP composition, was reflected in variations in HI content, TBARs values and texture parameters. The main differences between the two forms (C and F) can be attributed to the difference in the ratio of animal-source part to plant-source parts. 

In this study, N_2_-MAP limited the oxidation of the 3D-printed products and further stabilized their physicochemical characteristics and textural properties. Nevertheless, there was a dissociation in the animal and plant layers under this storage condition. Moreover, the measurements carried out allowed us to obtain results at the scale of the whole product, rather than layer by layer. A color change observed visually on the animal part is an indicator that can link biochemical and textural modifications to oxidation mechanisms. However, this observation is not sufficient for a precise characterization of the changes that occurred in each of the layers and at their interface.

Future studies are planned to determine iron distribution and oxidation state in both the animal part and the plant part, but also at their interface, in relation to time–course change in product texture during storage. Moreover, iron bioavailability in this kind of a hybrid product is a complex issue that can only be unraveled by integrating the dietary factors present. An approach at the microscopic scale could help to highlight key interactions between iron and other compounds. An evaluation of the taste of the products in relation to their acceptability by the consumers is also envisaged afterwards.

The iron-rich 3D-printed hybrid products developed here raise prospects for addressing iron deficiency, while at the same time adding value to certain animal by-products.

## Figures and Tables

**Figure 1 foods-12-01375-f001:**
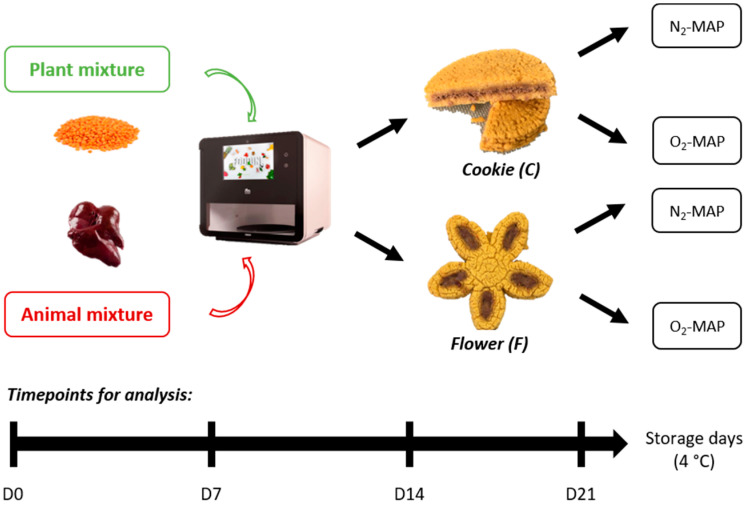
Schematic representation of the experiment illustrating the two printed forms studied (C and F) made from the two printable mixtures (plant mixture and animal mixture), their storage conditions (O_2_-MAP or N_2_-MAP; 4 °C), and the timepoints of the analyses performed (day 0 to 21). D: day.

**Figure 2 foods-12-01375-f002:**
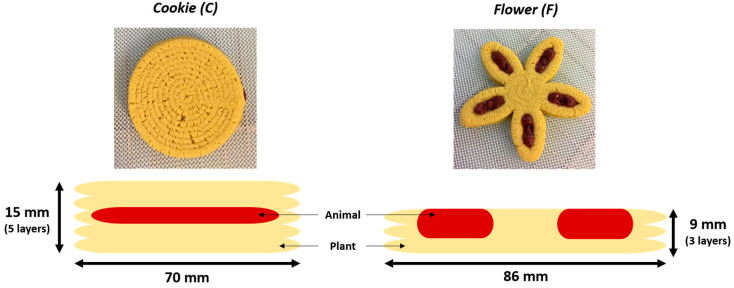
Detailed view of the 3D-printed food geometries and schematic distribution of animal-source (in red) and plant-source (in yellow) mixtures.

**Figure 3 foods-12-01375-f003:**
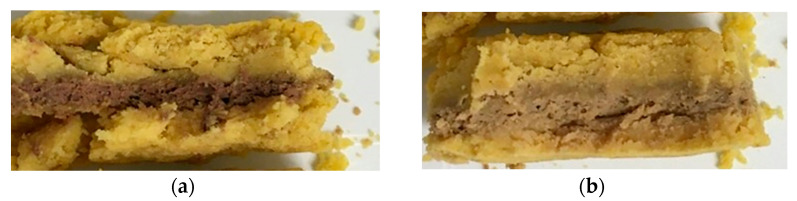
Photos of a longitudinal section of the 3D-printed food (form C) taken after 21 days in storage under (**a**) N_2_-MAP and (**b**) O_2_-MAP.

**Figure 4 foods-12-01375-f004:**
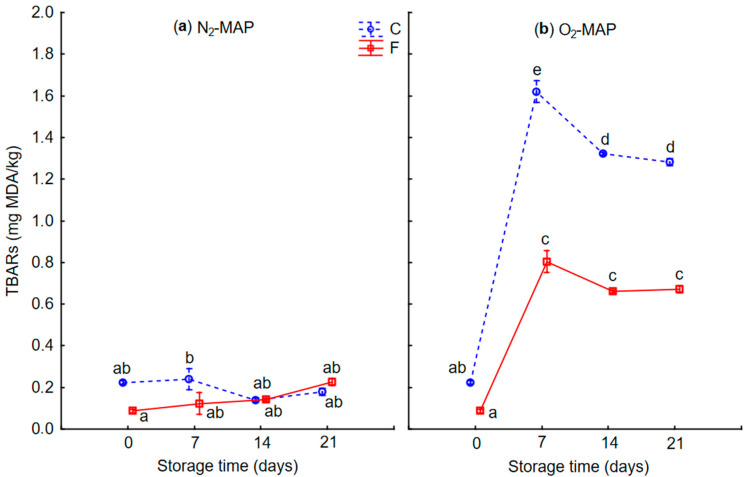
Time-course evolution of the TBARs values measured on 3D-printed products (forms C and F) as a function of MAP conditions (a) and (b), during storage for up to 21 days at 4 °C. N_2_-MAP (**a**): 70% nitrogen (N_2_) + 30% carbon dioxide (CO_2_); O_2_-MAP (**b**): 70% oxygen (O_2_) + 30% carbon dioxide (CO_2_). Lowercase superscript letters (a–e) indicate significant differences between samples according to form (C or F), MAP (O_2_-MAP or N_2_-MAP) and time in storage (*p* < 0.05). TBARs = ThioBarbituric Acid-Reactive substances. Each point is the mean of 3 replicates ± standard error of the mean (SEM).

**Figure 5 foods-12-01375-f005:**
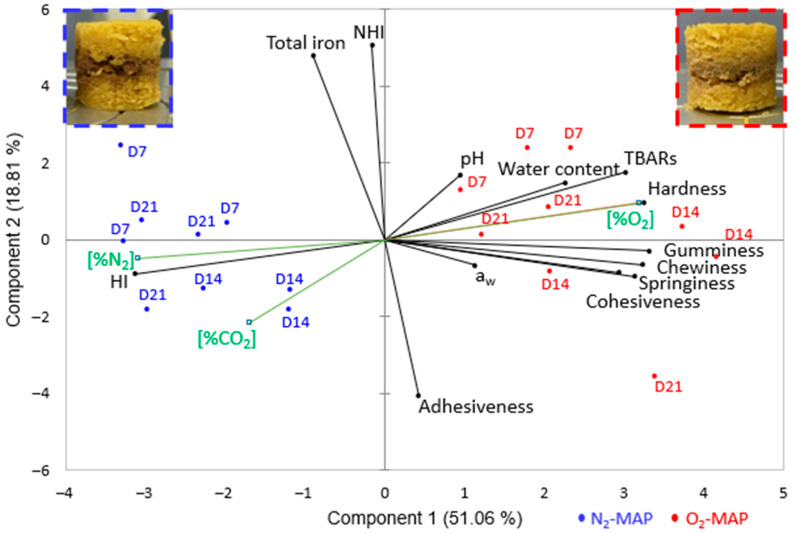
PCA map of 3D-printed form-C products showing the impact of time in storage and MAP conditions on physicochemical, nutritional and textural parameters. Black lines: principal variables; green lines: supplementary variables. Each trial is identified by time in storage (D7 to D21) and MAP (N_2_-MAP in blue; O_2_-MAP in red). Insets in the upper corners are representative of the samples obtained according to MAP conditions from day 7.

**Table 1 foods-12-01375-t001:** Composition and quantities of ingredients used for each part of the 3D-printed hybrid foods.

Animal Mixture		Plant Mixture	
Ingredients	Mass Fraction (%)	Ingredients	Mass Fraction (%)
Raw poultry liver	70.25	Red lentil puree	86.7
Raw pork liver	27.25	Lupin flour	9.7
Raspberry vinegar	2	Peanut oil	2.9
Salt	0.5	Curry powder	0.35
		Salt	0.35

**Table 2 foods-12-01375-t002:** Calibrated printing parameters used for the animal and plant mixtures.

Printing Parameters	Animal Part	Plant Part
Print speed (mm/min)	8000	8000
Ingredient flow speed	1.4	1.4
Line thickness (mm)	4.2	3.5
Distance between layers (mm)	3.5	3.5
Fill factor (0–2%)	1	0.9

**Table 3 foods-12-01375-t003:** Nutritional profile of printable food materials and 3D-printed food products estimated from compositional data and CIQUAL database (wet basis) calculations.

Sample	Energy (kcal/100 g)	Estimated Values (g/100 g)				
		Fat	Carbohydrate	Fiber	Protein	Salt
Animal mixture	159	5.7	1.0	0.0	25.2	0.7
Plant mixture	166	4.5	21.7	9.9	13.1	0.5
Form C (20:80 *)	165	4.8	17.6	7.9	15.5	0.5
Form F (15:85 *)	165	4.7	18.6	8.4	14.9	0.5

* Animal mixture/plant mixture ratio.

**Table 4 foods-12-01375-t004:** pH and water content in the two forms (C and F) of 3D-printed products stored for 21 days in the two modified atmospheres (N_2_-MAP and O_2_-MAP).

		pH		Water Content (kg Water/kg Dry Matter)	
	Days in Storage	N_2_-MAP	O_2_-MAP	N_2_-MAP	O_2_-MAP
Cookie (C)	0	6.28 ^a^ ± 0.00	6.28 ^a^ ± 0.00	1.63 ^ef^ ± 0.00	1.63 ^ef^ ± 0.00
	7	6.39 ^de^ ± 0.01	6.41 ^e^ ± 0.00	1.59 ^de^ ± 0.01	1.66 ^f^ ± 0.01
	14	6.39 ^cde^ ± 0.02	6.40 ^e^ ± 0.01	1.60 ^e^ ± 0.00	1.63 ^ef^ ± 0.01
	21	6.32 ^b^ ± 0.00	6.35 ^b^ ± 0.01	1.61 ^ef^ ± 0.00	1.62 ^ef^ ± 0.00
Flower (F)	0	6.41 ^e^ ± 0.00	6.41 ^e^ ± 0.00	1.48 ^a^ ± 0.01	1.48 ^a^ ± 0.01
	7	6.39 ^de^ ± 0.01	6.39 ^cde^ ± 0.01	1.48 ^a^ ± 0.01	1.49 ^ab^ ± 0.00
	14	6.42 ^e^ ± 0.00	6.40 ^e^ ± 0.00	1.55 ^cd^ ± 0.02	1.52 ^abc^ ± 0.01
	21	6.35 ^bc^ ± 0.01	6.36 ^bcd^ ± 0.01	1.54 ^bc^ ± 0.01	1.54 ^bc^ ± 0.00

Data are expressed as means ± SEM (*n* = 3). Lowercase superscript letters (a to f) for pH and water content according to MAP (O_2_-MAP and N_2_-MAP indicate significant differences between samples (*p* < 0.05)). O_2_-MAP with 70% oxygen (O_2_) + 30% carbon dioxide (CO_2_) and N_2_-MAP with 70% nitrogen (N_2_) + 30% carbon dioxide [CO_2_].

**Table 5 foods-12-01375-t005:** Mean contents of the different forms of iron measured on the raw materials and on the 3D-printed matrices.

Samples	HI (mg/100 g)	NHI (mg/100 g)	Total Iron (mg/100 g)
Chicken liver (raw)	5.79 ± 0.53	19.97 ± 1.13	25.77 ± 0.77
Pork liver (raw)	2.77 ± 1.10	16.82 ± 0.86	19.59 ± 1.81
Red lentils (cooked and mixed)	0.07 ± 0.00	8.69 ± 0.19	8.76 ± 0.19
Animal mixture	4.55 ± 0.08	17.11 ± 0.38	21.67 ± 0.31
Plant mixture	0.19 ± 0.00	9.14 ± 0.18	9.33 ± 0.18

Data are expressed as means ± SEM (*n* = 3) mg/100 g of the edible portion. Total iron (mg/100 g) = HI (mg/100 g) + NHI (mg/100 g).

**Table 6 foods-12-01375-t006:** Effect of form (‘cookie’ or ‘flower’), time in storage and MAP on the different forms of iron measured in the 3D-printed products form C.

	HI (mg/100 g)		NHI (mg/100 g)		Total Iron (mg/100 g)	
Days in Storage	N_2_-MAP	O_2_-MAP	N_2_-MAP	O_2_-MAP	N_2_-MAP	O_2_-MAP
0	0.98 ^c^ ± 0.01	0.98 ^c^ ± 0.01	15.41 ^b^ ± 0.72	15.41 ^b^ ± 0.72	16.40 ^b^ ± 0.72	16.40 ^b^ ± 0.72
7	0.92 ^c^ ± 0.02	0.33 ^a^ ± 0.00	14.41 ^ab^ ± 0.72	14.33 ^ab^ ± 0.41	15.32 ^ab^ ± 0.71	14.67 ^ab^ ± 0.41
14	0.81 ^b^ ± 0.02	0.36 ^a^ ± 0.00	12.31 ^a^ ± 0.16	13.17 ^ab^ ± 0.33	13.11 ^a^ ± 0.14	13.53 ^a^ ± 0.33
21	0.77 ^b^ ± 0.01	0.35 ^a^ ± 0.01	13.57 ^ab^ ± 0.47	13.49 ^ab^ ± 0.78	14.34 ^ab^ ± 0.48	13.84 ^ab^ ± 0.77

Data are expressed as means ± SEM (*n* = 3). Lowercase superscript letters (a to c) for different forms of iron (HI, NHI, total iron) according to MAP (O_2_-MAP and N_2_-MAP) indicate significant differences between samples (*p* < 0.05). O_2_-MAP with 70% oxygen (O_2_) + 30% carbon dioxide (CO_2_) and N_2_-MAP with 70% nitrogen (N_2_) + 30% carbon dioxide [CO_2_].

**Table 7 foods-12-01375-t007:** Effect of time in storage and MAP on the different forms of iron measured in the 3D-printed products form F.

	HI (mg/100 g)		NHI (mg/100 g)		Total Iron (mg/100 g)	
Days in Storage	N_2_-MAP	O_2_-MAP	N_2_-MAP	O_2_-MAP	N_2_-MAP	O_2_-MAP
0	0.90 ^c^ ± 0.01	0.90 ^c^ ± 0.01	14.86 ^c^ ± 0.23	14.86 ^c^ ± 0.23	15.77 ^c^ ± 0.22	15.77 ^c^ ± 0.22
7	0.79 ^b^ ± 0.04	0.44 ^a^ ± 0.00	13.42 ^bc^ ± 0.14	13.11 ^b^ ± 0.45	14.21 ^bc^ ± 0.18	13.54 ^b^ ± 0.44
14	0.72 ^b^ ± 0.02	0.42 ^a^ ± 0.00	10.81 ^a^ ± 0.33	12.43 ^ab^ ± 0.08	11.53 ^a^ ± 0.34	12.85 ^b^ ± 0.08
21	0.71 ^b^ ± 0.04	0.41 ^a^ ± 0.01	11.77 ^ab^ ± 0.79	11.64 ^ab^ ± 0.55	12.48 ^ab^ ± 0.82	12.05 ^ab^ ± 0.54

Data are expressed as means ± SEM (*n* = 3). Lowercase superscript letters (a to c) for different forms of iron (HI, NHI, total iron) according to MAP (O_2_-MAP and N_2_-MAP) indicate significant differences between samples (*p* < 0.05). O_2_-MAP with 70% oxygen (O_2_) + 30% carbon dioxide (CO_2_) and N_2_-MAP with 70% nitrogen (N_2_) + 30% carbon dioxide [CO_2_].

**Table 8 foods-12-01375-t008:** Effect of time in storage and MAP conditions on the textural properties measured on the 3D-printed food form C.

MAP	Days in Storage	Hardness (N)	Springiness	Cohesiveness	Gumminess (N)	Chewiness (N)
N_2_-MAP	0	4.93 ^a^ ± 0.11	0.12 ^a^ ± 0.01	0.14 ^a^ ± 0.02	0.71 ^a^ ± 0.10	0.09 ^a^ ± 0.02
	7	7.31 ^b^ ± 0.06	0.14 ^a^ ± 0.01	0.12 ^a^ ± 0.01	0.89 ^ab^ ± 0.05	0.13 ^a^ ± 0.01
	14	7.63 ^b^ ± 0.15	0.17 ^ab^ ± 0.02	0.13 ^a^ ± 0.00	0.97 ^ab^ ± 0.02	0.16 ^ab^ ± 0.02
	21	7.24 ^b^ ± 0.18	0.13 ^a^ ± 0.01	0.12 ^a^ ± 0.00	0.88 ^ab^ ± 0.04	0.11 ^a^ ± 0.01
O_2_-MAP	0	4.93 ^a^ ± 0.11	0.12 ^a^ ± 0.01	0.14 ^a^ ± 0.02	0.71 ^a^ ± 0.10	0.09 ^a^ ± 0.02
	7	9.12 ^c^ ± 0.42	0.17 ^ab^ ± 0.01	0.13 ^a^ ± 0.00	1.19 ^bc^ ± 0.08	0.21 ^abc^ ± 0.03
	14	9.27 ^c^ ± 0.29	0.21 ^b^ ± 0.01	0.16 ^a^ ± 0.00	1.50 ^c^ ± 0.09	0.32^c^ ± 0.03
	21	8.93 ^c^ ± 0.20	0.18 ^ab^ ± 0.02	0.15 ^a^ ± 0.00	1.38 ^c^ ± 0.06	0.25 ^bc^ ± 0.03

Data are expressed as means ± SEM (*n* = 3). Lowercase superscript letters (a to c) in the same column indicate significant differences between samples (*p* < 0.05). O_2_-MAP with 70% oxygen [O_2_] + 30% carbon dioxide (CO_2_) and N_2_-MAP with 70% nitrogen (N_2_) + 30% carbon dioxide [CO_2_].

## Data Availability

The data presented in this study are available on request from the corresponding author.

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
