# Peer review of "Investigation into the Physicochemical and Textural Properties of an Iron-Rich 3D-Printed Hybrid Food"

_foods, 2023, doi:10.3390/foods12071375_

Round 1

Reviewer 1 Report

Investigation into the physicochemical and textural properties of an iron-rich 3D-printed hybrid food

This study of Investigation into the physicochemical and textural properties of an iron-rich 3D-printed hybrid food provide new insight in producing product based on the combination of various edible material. The methodology was properly design and specific. Therefore, I would suggest this paper to be published in Foods, subject to the following amendment;

Reviewer’s Comments

1.     Line 64 – 65 – “This type of food is in line with current consumption trends and 64 nutritional recommendations to rebalance the ratio of animal protein to plant protein to-65 wards more plant protein” – need a reference

2.     Is there any specific reason for using animal liver (poultry and pork) in this study?

3.     The methodology was sufficient in this study.

4.     It is suggested that the author should improve the resolution of Figure 3

5.     Table 6 – Iron results. It is suggested that the researcher revise the result carefully as In my opinion, there are doubtful results were obtained. For instances. 14.21abcdef ± 0.18 and 11.53a ± 0.34. How can this be considered not significant (based on the superscript)?

6.     There is an inconsistency of all the Tables format (results)

7.     Is there any analysis to justify the colour changes of the animal part in within the printed product? (Figure 7)

8.     It is suggested that the sentence in Line 566 – can be removed from the manuscript. 

Reviewer 2 Report

This manuscript (MS) investigates the feasibility of formulating an iron-rich hybrid food product blending plant-source and animal-source proteins ingredients for iron-deficient populations. As we all known, anemia, which is a global health threat, mainly affects preschool-aged children and pregnant women, but is also associated with a sedentary lifestyle and is more prevalence in developing nations. The MS is interesting and provides helpful information for developing hybrid food products that upvalue animal by-products. The language is good. I suggest it can be accepted after minor revision. Some comments are listed below.

1. The 3D-printed products in this MS were made from pork and chicken liver and red lentils. It definitely has some useful for anemia as liver contains high content of iron. However, the author did not evaluate the taste of the products, as this is very important for the acceptance by consumers. Whether the food made from the animal liver and plant protein has good taste and flavor ? In addition, the author just mentioned that the products were prepared by baking. Whether the products can be accepted by consumers ?

2. Line 184-185, how to measure the core temperature by K-type thermocouple. Afer baking, the products may become very hard. I think it is very different to insert the probe of thermocouple to the core of the products.

3. The authors should revise the equations. Please note the subscript specification of the letters.

4. For the Fig. 5, why the authors just shows the results of PCA analysis of form-C products. How about form-F ? Additionally, the total variance for principal components 1 and 2 is only 69.9%. Whether the analysis is credible as the value (69.9%) is low.
